# TAMP: TASK-AWARE MULTIMODAL PRE-INTERACTION FOR FINE-GRAINED LARGE LANGUAGE MODELS

## ABSTRACT

Current Multimodal Large Language Models (MLLMs) primarily rely on image-level visual-linguistic alignment, limiting their capability in fine-grained visual perception tasks. Existing solutions either serialize coordinates as text inputs, which lose spatial semantics, or introduce specialized expert modules that increase inference latency and exhibit task bias. To address these limitations, we propose TAMP, a Task-aware Multimodal Pre-Interaction for Fine-Grained Multi-modal LLMs, that automatically recognizes key task-relevant information from instructions and extracts corresponding region features through a unified and detector-free paradigm. A task-aware region connector with a dual-branch is designed that dynamically handles both referring and grounding tasks. By introducing an instruction template with region placeholders, we seamlessly integrate fine-grained region features into the LLM's reasoning process. Extensive experiments demonstrate that our approach achieves state-of-the-art performance on both referring and grounding benchmarks while maintaining strong general VQA capabilities.

## 1 INTRODUCTION

Multi-modal Large Language Models (Alayrac et al., 2022; Dai et al., 2023; Liu et al., 2023b; Zhu et al., 2023) (MLLMs) have achieved remarkable progress in visual-language understanding tasks. However, current MLLMs still struggle with fine-grained visual perception tasks such as referring (Mao et al., 2016; Yu et al., 2016) and grounding (Mao et al., 2016) tasks. The main limitation is that mainstream approaches primarily align image-level visual features with LLMs through multimodal instruction tuning based on image-text pairs, while lacking region-level multimodal alignment and supervision signals. This coarse-grained alignment prevents them from accurately localizing objects and modeling spatial relationships in complex scenes. To address the limitations of MLLMs on fine-grained visual perception tasks, existing research has proposed three main solution paradigms, as illustrated in Figure 1.

Early works (Zhao et al., 2023; Chen et al., 2024b) attempted to feed regions of interest as serialized bounding box coordinates as text prompts, enabling region-level modeling without modifying the model architecture as shown in Figure 1(a). However, LLMs are inherently adept at processing discrete symbols but lack capabilities for modeling continuous spatial coordinates. This causes spatial semantics to be easily lost during the encoding process, making it challenging to efficiently integrate with visual cues and linguistic context. Subsequent work (Pi et al., 2024; You et al., 2023) shifted to using high-dimensional vectors to represent positional information via incorporating specialized expert modules at MLLMs' input or output as shown as 1(b)(c). However, this paradigm suffers from task bias and efficiency issues: input-side modules benefit referring but not grounding, while output-side modules enhance grounding but require additional processing steps, increasing latency. Multiple loss functions also complicate optimization with limited referring improvements. To jointly address the performance bottlenecks of referring and grounding, latest methods (Ma et al., 2024; Yin et al., 2025) rely on pretrained object detectors (Zhu et al., 2020) for region

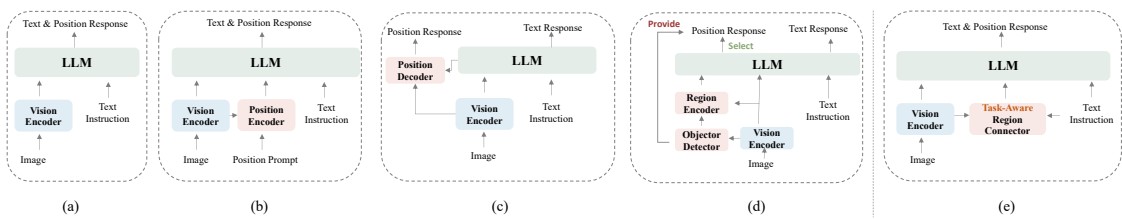

Figure 1: (a) Direct serializing spatial coordinates as text tokens; (b) External input-side position encoder for referring; (c) External output-side position decoder for grounding; (d) External object detector providing region proposals; (e) Task-aware region encoder for unified referring and grounding (Ours).

proposals, extracting local features via mechanisms like ROIAlign (He et al., 2017) for fine-grained reasoning, as illustrated in Figure 1(d). While leveraging mature detection techniques, this approach faces inherent limitations. The performance ceiling of object detectors and domain shift issues fundamentally constrain the model's localization accuracy. Furthermore, numerous task-irrelevant candidate features significantly extend sequence length, increasing computational complexity while introducing additional noise. In fact, the instructions in fine-grained visual tasks contain heuristic clues that could guide the model to selectively attend to task-relevant key regions.

In this paper, we propose a novel Task-aware Multimodal Pre-Interaction Framework, dubbed TAMP, that can enhance region representation prior to the LLM reasoning. By introducing the pre-perceived region features into the large language model, it can explicitly guide the model to focus on task-relevant region, thereby achieving more precise region-level multimodal understanding, as illustrated in Figure 1(e). Specifically, TAMP employs a task-aware region connector which uniformly handles referring and grounding fine-grained visual tasks through a dual-branch architecture. This connector first automatically parses the task type and semantic information of the input instruction through a task extractor, then dynamically activates the corresponding processing branch and extracts task-relevant fine-grained region features. Furthermore, we design an instruction template with a region placeholder <region> , which is dynamically replaced with task-aware region feature, enabling seamless integration of region features into text instructions for subsequent multimodal reasoning.

Our main contributions are summarized as follows:

- We propose a novel Task-aware Multimodal Pre-Interaction Framework for LLM. It explicitly enhances region representation prior to LLM reasoning by performing pre-interaction between image features and task-relevant instruction embeddings, which explicitly guide the model to focus on task-critical regions.

- We design the Task-Aware Region Connector with a unified dual-branch structure to distill task-specific saliency cues and key information from fine-grained instructions. A unified and detector-free training paradigm is achieved by introducing a region placeholder, enabling LLM-friendly instruction tuning while preserving spatial semantics.

- Extensive experiments demonstrate that our approach achieves state-of-the-art performance on both referring and grounding benchmarks, surpassing all existing MLLMs, and maintains robust image-level understanding and reasoning capabilities on traditional general VQA benchmarks.

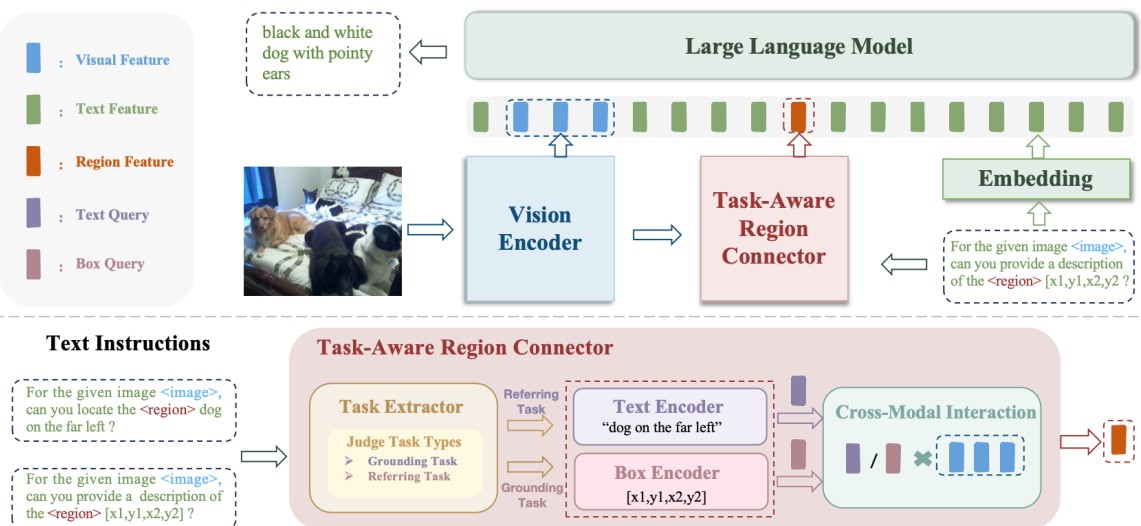

Figure 2: Overview of TAMP. We propose a TAMP Framework for fine-grained visual perception, which is a unified and detector-free training paradigm. The Task-Aware Region Connector is able to distill task-specific saliency cues and key information from instructions for the regions.

## 2 METHODS

### 2.1 OVERVIEW

Despite the remarkable progress MLLMs have made in general visual understanding, existing methods exhibit two critical limitations when handling fine-grained visual tasks. First, current approaches fail to fully leverage task-specific information in instructions; Second, these methods lack the capability to selectively focus on relevant visual regions based on task requirements, instead processing the entire image uniformly. Our key insight is that fine-grained visual tasks inherently contain rich task-specific information within their instructions-natural language descriptions for referring tasks and spatial coordinates for grounding tasks, which should guide the model to selectively attend to relevant visual regions. Based on this, we propose a task-aware region connector that establishes effective interaction between task-aware query which encode task-specific information extracted from instructions, and visual features through a unified dual-branch architecture, as illustrated in Figure 2. This module elegantly handles both referring and grounding fine-grained tasks within a unified framework.

### 2.2 MODEL ARCHITECTURE

As illustrated in Figure 2, TAMP comprises four core components: (1) a vision encoder for image tokenization; (2) a task-aware region connector that dynamically extracts fine-grained region features conditioned on the key task information; (3) visual/region projectors that project visual and region features into the language space; and (4) a LLM for unified modeling of multimodal inputs and outputs.

**Vision Encoder.** We employ a pre-trained Vision Transformer (ViT) (Dosovitskiy et al., 2020) as our vision encoder $\Phi_V$. This encoder is pre-trained through image-text contrastive learning, enabling its patch-level visual features to capture comprehensive global semantic representations. For an input image $I \in \mathbb{R}^{3 \times W \times H}$, the encoder $\Phi_V$ produces a sequence of visual features $F_V = \Phi_V(I)$.

**Task-aware Region Connector.** Our task-aware region connector $\Phi_R$ enables task-aware fine-grained region feature extraction through three key components: (1) a task extractor that parses task-specific information from instructions; (2) a dual-branch encoder to generate task-aware query based on task information; and (3) a cross-modal interaction module dynamically aggregates task-relevant fine-grained region features. To facilitate effective task information extraction, we design specific task templates for both grounding and referring tasks. For referring tasks, the task extractor extracts natural language descriptions within $<ref>$ tags and converts them into a semantic query vector $Q_{text}$ through a pre-trained BERT encoder (Devlin et al., 2019). The query vector encodes the semantic information of the target region description; For grounding tasks, the task extractor identifies normalized bounding box coordinates $[x_1, y_1, x_2, y_2]$ within $<loc>$ tags and generates position-aware query $Q_{box}$ through a multi-layer MLP to precisely localize spatial regions. Both types of queries are transformed to a shared query space through a unified projection layer, generating the task-aware query $Q$ that serves as input to the cross-modal interaction module. This module employs a query-driven attention mechanism that dynamically adjusts the aggregation strategy of visual features based on task type:

$$\hat{F}_R = \text{CrossAttn}(Q, F_V) = \text{softmax}\left(\frac{QW_Q(F_V W_K)^T}{\sqrt{D_k}}\right) \cdot F_V W_V, \tag{1}$$

$$F_R = \text{LayerNorm}(\text{FFN}(\hat{F}_R)), \tag{2}$$

where $F_V$ represents the feature map output by the visual encoder, $W_V$, $W_Q$, and $W_K$ are learnable projection matrices. The detailed instruction templates for specific tasks are provided in Appendix D.

**Visual/Region Projector.** We utilize two parallel MLP projection heads to transform visual features into the language space: the Visual Projector $P_V$ maps the global visual representation $F_V$ to the visual global token $H_v$, while the Region Projector $P_R$ maps fine-grained region features $F_R$ to region tokens $H_r$. This dual-path projection ensures that different types of visual information are processed independently and aligned within a unified language representation space, thereby preserving individualized semantics while enables effective fusion with textual embeddings.

**LLM.** We adopt the pre-trained LLaMA (Touvron et al., 2023) as our language model and keep its original parameter initialization. The language instruction is embedded by the LLM token embedding layer, which produces language tokens $H_t$. The LLM takes as input a concatenation of three sequences: the text instruction tokens, global visual tokens produced by the Visual Projector, and region tokens produced by the Region Projector. The probability p of the next token at position i is computed as follows:

$$p(X_a|H_t, H_v, H_r) = \prod_{i=1}^{L} p(x_i|H_t, H_v, H_r, x_{<i}) \tag{3}$$

where $X_a$ represents the generated answer tokens, $x_{<i}$ represents all previously generated tokens before position $i$, $L$ is the length of the generated sequence.

## 2.3 TASK-AWARE INSTRUCTION TEMPLATE DESIGN

To seamlessly integrate fine-grained spatial perception into multimodal large language models, we propose a unified, task-aware instruction template framework. This framework introduces special region tokens and replaces them with task-specific fine-grained region features prior to inference, enabling the model to address a variety of fine-grained visual perception tasks within a single unified paradigm. Current mainstream MLLMs typically adopt standard image–text template for instruction tuning. While this design is suitable for image-level vision-language tasks (e.g., Image Caption and VQA), it cannot effectively support fine-grained tasks that require region-level understanding. To address this limitation, we extend the standard template by introducing a Region Placeholder $<region>$ which is a special token that will be dynamically replaced with

task-relevant region features extracted by the Task-Aware Region Connector during the model's forward propagation. The comparison between standard and fine-grained template is shown below:

> **Standard Template**
>
> **Image:** {Image Tokens}
> **User:** {Task Instruction}
> **Assistant:** {Response}

> **Fine-grained Template**
>
> **Image:** {Image Tokens}
> **Region:** {<region>}
> **User:** {Task Instruction}
> **Assistant:** {Response}

Figure 3: Standard and fine-grained templates.

### 2.4 MODEL TRAINING STRATEGY

We adopt a three-stage progressive training strategy to address key bottlenecks in existing region-level methods. Latest works rely on pre-trained object detectors and operations like ROIAlign for region feature extraction. This "detect-extract-aggregate" pipeline has fundamental issues: region features from object detectors are not aligned with language space, and ROI-aggregated features have limited text-semantic alignment, increasing inference latency and impairing downstream task performance.

Our strategy progressively establishes visual-language alignment from region-level to image-level. First, We train the task-aware region connector to directly establish region-level visual-language alignment. For referring tasks, we adopt semantic contrastive loss to ensure accurate capture of region visual content and mapping to the language space. For grounding tasks, we combine the same semantic contrastive loss with localization loss to learn both text-region semantic alignment and spatial localization capabilities. Then, We train the projection layer using large-scale image-text pairs while freezing the task-aware region connector and language model, establishing image-level visual-language alignment. Finally, we simultaneously optimize the projection layer, task-aware region connector, and region projection layer, while fine-tuning the visual encoder and LLM through LoRA, enabling comprehensive multimodal understanding and instruction-following capabilities. Details of the training datasets are provided in Appendix C.

## 3 EXPERIMENTS

### 3.1 EXPERIMENTAL SETTING

We implement two variants of TAMP, employing CLIP ViT-L-224px(Radford et al., 2021) and EVA-G-224px (Sun et al., 2023) as image backbones respectively, with LLaMA-2-7B(Touvron et al., 2023) serving as the LLM. For efficient training, we perform parameter-efficient fine-tuning on both the visual encoder and LLM through LoRA(Hu et al., 2022), and insert lightweight modules before each self-attention layer. Our baseline follows the same architecture but without the task-aware region connector, using only global visual features like conventional MLLM LLaVA (Liu et al., 2023b). Detailed training configurations are provided in Appendix C.

### 3.2 GROUNDING BENCHMARK RESULTS

We evaluate TAMP on RefCOCO (Kazemzadeh et al., 2014), RefCOCO+ (Yu et al., 2016), and RefCOCOg (Mao et al., 2016) benchmarks to assess its localization capability. As shown in Table 1, TAMP significantly outperforms baselines in both zero-shot (63.61% with EVA-G) and fine-tuning (88.20% with EVA-G) settings, surpassing all existing models comparable or larger scale, as well as existing specialized grounding

Table 1: Comparison on Grounding Benchmark. "Avg." means the average of top-1 accuracy over all the 8 evaluation sets.

| Method | Model type | RefCOCO | | | RefCOCO+ | | | RefCOCOg | | Avg. |
|---|---|---|---|---|---|---|---|---|---|---|
| | | val | testA | testB | val | testA | testB | val | test | |
| MDETR (Kamath et al., 2021) | | 86.75 | 89.58 | 81.41 | 79.52 | 84.09 | 70.62 | 81.64 | 80.89 | 81.81 |
| G-DINO (Liu et al., 2024) | Specialist | 90.56 | 93.19 | 88.24 | 82.75 | 88.95 | 75.92 | 86.13 | 87.02 | 86.60 |
| UNINEXT-L (Yan et al., 2023) | | 91.43 | 93.73 | 88.93 | 83.09 | 87.90 | 76.15 | 86.91 | 87.48 | 86.95 |
| *Zero-shot Setting* | | | | | | | | | | |
| Kosmos-2 (Peng et al., 2023) | | 52.32 | 57.42 | 47.26 | 45.48 | **50.73** | 42.24 | 60.57 | 61.65 | 52.21 |
| GRILL (Jin et al., 2023) | | - | - | - | - | - | - | - | 47.50 | - |
| Pink (Xuan et al., 2024) | | 54.10 | 61.20 | 44.20 | 43.90 | 50.70 | 35.00 | 59.10 | 60.10 | 51.00 |
| LION-12B (Chen et al., 2024a) | Generalist | 58.54 | 56.41 | 59.36 | 45.93 | 45.73 | 47.89 | 66.12 | 64.69 | 55.58 |
| BaseLine-CLIP-L | | 56.06 | 63.21 | 47.11 | 45.87 | 51.03 | 38.92 | 62.25 | 60.45 | 53.11 |
| TAMP-CLIP-L | | 63.21 | 67.85 | 56.86 | 50.24 | 54.63 | 44.61 | 68.42 | 66.59 | 59.05 |
| BaseLine-EVA-G | | 62.41 | 66.50 | 58.63 | 49.35 | 54.44 | 46.49 | 68.59 | 67.61 | 59.25 |
| TAMP-EVA-G | | **68.16** | **69.98** | **64.14** | **54.03** | **56.44** | **51.67** | **72.63** | **71.83** | **63.61** |
| *Fine-tuning Setting* | | | | | | | | | | |
| VisionLLM-H (Wang et al., 2023) | | - | 86.70 | - | - | - | - | - | - | - |
| Shikra-13B (Chen et al., 2023c) | | 87.83 | 91.11 | 81.81 | 82.89 | 87.79 | 74.41 | 82.64 | 83.16 | 83.96 |
| GroundingGPT (Li et al., 2024) | | 88.02 | 91.55 | 82.47 | 81.61 | 87.18 | 73.18 | 81.67 | 81.99 | 83.46 |
| Ferret-13B (You et al., 2023) | | 89.48 | 92.41 | 84.36 | 82.81 | 88.14 | 75.17 | 85.83 | 86.34 | 85.57 |
| Pink (Xuan et al., 2024) | | 88.30 | 91.70 | 84.00 | 81.40 | 87.50 | 73.70 | 83.70 | 83.70 | 84.25 |
| MiniGPTv2 (Chen et al., 2023b) | | 88.69 | 91.65 | 85.33 | 79.97 | 85.12 | 74.45 | 84.44 | 84.66 | 84.29 |
| PerceptionGPT-13B (Pi et al., 2024) | | 89.17 | 93.20 | 85.96 | 83.72 | 89.19 | 75.31 | 83.75 | 84.69 | 85.62 |
| Qwen-VL (Bai et al., 2023) | | 89.36 | 92.26 | 85.34 | 83.12 | 88.25 | 77.21 | 85.58 | 85.48 | 85.83 |
| LION-12B (Chen et al., 2024a) | | 89.80 | 93.02 | 85.57 | 83.95 | 89.22 | 78.06 | 85.52 | 85.74 | 86.36 |
| VPP-LLaVA-13B (Tang et al., 2025) | Generalist | 90.32 | 93.02 | 86.34 | 84.65 | **90.78** | 79.06 | 85.64 | 86.01 | 86.98 |
| Groma (Ma et al., 2024) | | 89.53 | 92.09 | 86.26 | 83.90 | 88.91 | 78.05 | 86.37 | 87.01 | 86.52 |
| ROD-MLLM (Yin et al., 2025) | | 90.2 | 93.0 | 86.3 | 84.8 | 89.9 | 77.5 | 86.7 | 86.7 | 86.89 |
| BaseLine-CLIP-L | | 88.11 | 91.87 | 82.87 | 81.15 | 87.50 | 72.57 | 82.43 | 83.44 | 83.74 |
| TAMP-CLIP-L | | 89.28 | 92.15 | 84.06 | 82.77 | 88.18 | 74.78 | 84.17 | 85.46 | 85.11 |
| BaseLine-EVA-G | | 91.55 | 92.89 | 88.07 | 84.68 | 89.10 | 79.40 | 86.13 | 86.78 | 87.33 |
| TAMP-EVA-G | | **91.76** | **93.92** | **88.34** | **86.30** | 90.13 | **79.93** | **86.99** | **88.21** | **88.20** |

methods. Notably, while methods like LION, Ferret, and Groma require millions of grounding samples, TAMP achieves superior performance using only 0.5M data through our lightweight task-aware region connector. This fully validates the effectiveness of the task-aware multimodal pre-interaction paradigm.

## 3.3 REFERRING BENCHMARK RESULTS

We evaluated TAMP's fine-grained region understanding capability on the RefCOCOg(Mao et al., 2016) and Visual Genome datasets(Krishna et al., 2017). We employed METEOR and CIDEr as evaluation metrics, which comprehensively measure both the accuracy and fluency of generated descriptions. As shown in Table 2, TAMP-EVA-G achieved the best performance on both datasets, attaining 17.7 METEOR and 117.3 CIDEr on RefCOCOg, and 19.4 METEOR and 164.0 CIDEr on Visual Genome. Particularly noteworthy is that compared to the corresponding baseline variants, TAMP demonstrated consistent performance improvements across both visual encoder architectures.

### 3.4 GENERAL VQA BENCHMARK RESULTS

To verify whether our model maintains general visual question answering capabilities while enhancing fine-grained visual understanding abilities, we conducted evaluations on five widely adopted VQA benchmarks: VQAv2(Goyal et al., 2017), AOK-VQA(**?**), VSR(Liu et al., 2023a), OK-VQA(Marino et al., 2019), and GQA(Hudson & Manning, 2019). These datasets comprehensively examine the model's multimodal understanding capabilities from different dimensions.

Table 2: Comparison with MLLMs on Referring Benchmark. $^\dagger$ indicates an extra stage of task-specific supervised tuning.

| Method | RefCOCOg | | Visual Genome | |
| --- | --- | --- | --- | --- |
| | METEOR | CIDEr | METEOR | CIDEr |
| GRIT (Wu et al., 2024) | 15.2 | 71.6 | 17.1 | 142 |
| Kosmos-2 (Peng et al., 2023) | 14.1 | 62.3 | - | - |
| LLaVA-v1.5-7B (Liu et al., 2023b) | 12.0 | 73.1 | - | - |
| VPP-LLaVA-7B (Tang et al., 2025) | 12.1 | 73.1 | - | - |
| ChatterBox (Tian et al., 2024) | 14.5 | - | - | - |
| GPT4RoI (Zhang et al., 2024b) | - | - | 17.4 | 145.2 |
| RegionGPT (Guo et al., 2024) | 16.9 | 109.9 | 17.0 | 145.6 |
| GLaMM$^\dagger$ (Rasheed et al., 2024) | 16.1 | 101.9 | 19.0 | 163.9 |
| Groma (Ma et al., 2024) | 16.8 | 107.3 | 19.0 | 158.4 |
| ROD-MLLM (Yin et al., 2025) | 17.3 | 113.8 | 19.0 | 158.5 |
| BaseLine-CLIP-L | 16.1 | 106.3 | 17.9 | 143.9 |
| TAMP-CLIP-L | 17.0 | 110.8 | 18.5 | 158.2 |
| BaseLine-EVA-G | 17.2 | 113.9 | 17.9 | 140.1 |
| TAMP-EVA-G | **17.7** | **117.9** | **19.4** | **164.0** |

As shown in Table 3, our method achieves comparable or superior performance to baseline on most benchmarks, indicating that we have not compromised the model's general visual understanding capabilities while introducing fine-grained perception abilities. Notably, despite using only 595K pretraining data and 797K instruction tuning data, we still achieve comparable performance to general models trained on larger-scale datasets on most traditional VQA benchmarks. This result strongly demonstrates that our proposed task-aware multimodal pre-interaction paradigm not only effectively enhances fine-grained perception capabilities but also successfully maintains the model's general visual understanding abilities, achieving a good balance between fine-grained understanding and general capabilities.

### 3.5 ABLATION STUDY

Table 4: Referring and Grounding abilities with different box encoder designs.

| Box Encoder | Referring | Grounding |
| --- | --- | --- |
| Multi-Layer MLP | 110.8 | 85.11 |
| Sin/Cos Encoding | 101.6 | 84.95 |
| Sin/Cos + Linear | 105.0 | 84.91 |

Table 5: Ablation Study on Freezing the task-aware region connector.

| Status | Referring | Grounding |
| --- | --- | --- |
| frozen | 84.74 | 110.0 |
| unfrozen | 85.11 | 110.8 |

**Impact of Box Encoder Design.** Table 4 presents ablation results for different bounding box encoder designs in TAMP-CLIP-L, we measure referring ability with CIDEr score on Refcocog and grounding ability

Table 3: Comparison with MLLMs on General VQA Benchmark. *indicates that the training data does not include the GQA dataset. [†] means our evaluated results by using publicly released checkpoints.

| Models | Vision Encoder | Resolution | Training Data | | Performance | | | | |
|---|---|---|---|---|---|---|---|---|---|
| | | | #PT Data | #IT Data | VQAv2 | AOK-VQA | VSR | OK-VQA | GQA |
| InstructBLIP-7B(Dai et al., 2023) | EVA-G | 224 | 129M | 1.2M | - | - | 54.3 | - | 49.2 |
| Shikra-7B (Chen et al., 2023c) | ViT-L | 224 | 595K | 5.5M | 76.7 | - | 63.3 | 53.5 | 47.4* |
| Qwen-VL-7B (Bai et al., 2023) | ViT-G | 448 | 1.4B | 50M | 78.8 | - | - | 58.6 | 59.3 |
| LLaVA-1.5-7B (Liu et al., 2023b) | ViT-L | 336 | 558K | 665K | 78.5 | - | 67.6 | - | **62.0** |
| MiniGPT-4-13B (Zhu et al., 2023) | EVA-G | 224 | 5M | 3.5K | - | - | 41.6 | 37.5 | 30.8* |
| MiniGPTv2-7B (Chen et al., 2023b) | EVA-G | 448 | >5M | >20M | - | - | 60.6 | 56.9 | 60.3 |
| Pink-7B[†] (Xuan et al., 2024) | ViT-L | 224 | 595K | 1.72M | - | 78.54 | 66.12 | 58.46 | 54.74* |
| Lions-4B (Chen et al., 2024a) | EVA-G | 224 | 2.7M | 9.9M | - | 59.98 | 72.96 | 51.08 | 49.50* |
| Lions-13B (Chen et al., 2024a) | EVA-G | 224 | 2.7M | 9.9M | - | **60.87** | **73.77** | 57.33 | 51.56* |
| BaseLine-7B | EVA-G | 224 | 595K | 797K | 79.08 | 58.94 | 69.22 | 59.84 | 54.85* |
| TAMP-7B | EVA-G | 224 | 595K | 797K | **79.84** | 59.64 | 68.49 | **59.96** | 55.89* |

Table 6: Performance comparison with different visual encoders and language models on the Grounding benchmarks.

| Model | Vision Encoder(Res) | LLM | RefCOCO | | | RefCOCO+ | | | RefCOCOg | | Avg. |
|---|---|---|---|---|---|---|---|---|---|---|---|
| | | | val | testA | testB | val | testA | testB | val | test | |
| Baseline | CLIP-L/14(336px) | LLaMA-2-7B | 89.93 | 92.61 | 84.88 | 82.60 | 89.21 | 74.15 | 85.07 | 85.93 | 85.55 |
| Ours | CLIP-L/14(336px) | LLaMA-2-7B | **90.31** | **93.64** | **85.56** | **84.05** | **90.31** | **76.68** | **86.25** | **86.43** | **86.65** |
| Baseline | CLIP-L/14(336px) | Vicuna1.5-7B | 89.98 | 93.11 | 84.67 | 83.65 | 89.61 | 75.25 | 85.44 | 86.23 | 85.99 |
| Ours | CLIP-L/14(336px) | Vicuna1.5-7B | **90.67** | **93.21** | **85.55** | **84.43** | **89.98** | **76.52** | **86.13** | **86.57** | **86.63** |

with average accuracy on grounding benchmarks. As we adopt a dual-branch architecture, the box encoder design primarily affects referring tasks that require processing spatial coordinates, while grounding tasks rely on the text branch for encoding linguistic descriptions. Compared to sinusoidal/cosine encoding (score 105.0) and its combination with linear layers (score 105.9), the multi-layer MLP achieves the best referring accuracy (score 110.8). This advantage stems from the MLP's ability to learn complex nonlinear mappings from normalized coordinates to the shared query space. While sinusoidal positional encoding effectively captures absolute positions, it has limited expressiveness in transforming spatial coordinates into task-aware queries that can interact with semantic visual features.

**Frozen task-aware region connector** As shown in Table 5, we freeze the task-aware region connector after training to evaluate its effectiveness and we measure referring ability with CIDEr score on Refcocog and grounding ability with average accuracy on grounding benchmarks. The frozen module maintains strong performance with 84.74% on Referring and 110.0 CIDEr on grounding tasks, experiencing minimal drops of 0.37% and 0.8 points respectively. Notably, even with frozen parameters, our model substantially outperforms baseline grounding benchmarks. This demonstrates the effectiveness of our first-stage pretraining, where we specifically optimize the task-aware region connector to establish robust region-level visual-language alignment through semantic contrastive and localization losses, ensuring our proposed task-aware region connector effectively capture task-relevant region features.

**Impact of Vision and Language Backbones.** Table 6 demonstrates the generalizability of our approach across different backbone architectures. When upgrading from CLIP-L/14(224px) (Radford et al., 2021) to CLIP-L/14(336px) (Radford et al., 2021) with LLaMA-2-7B (Touvron et al., 2023), our method achieves 86.65% average accuracy compared to the baseline's 85.55%, maintaining a consistent improvement margin

across different visual resolutions. Furthermore, when switching the language model from LLaMA-2-7B to Vicuna1.5-7B (Chiang et al., 2023) while maintaining CLIP-L/14(336px), our approach yields 86.63% compared to 85.99% baseline, with consistent improvements across all RefCOCO splits. These results validate that our task-aware region connector effectively enhances region-level understanding across different vision encoder resolutions and language model architectures, confirming the robustness and transferability of our design.

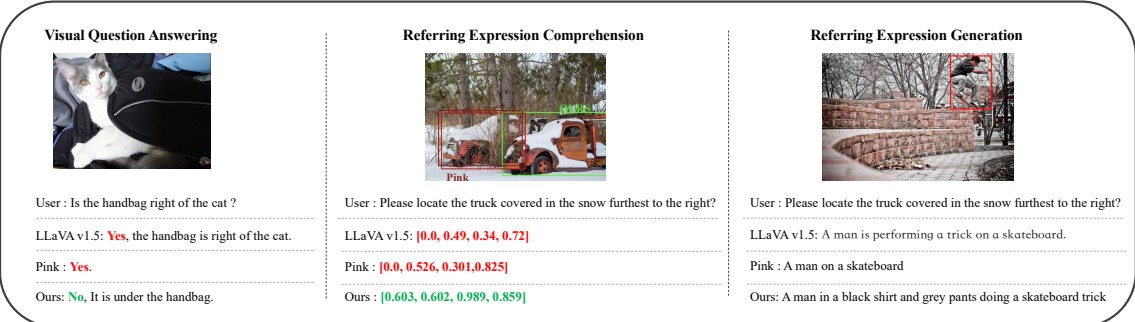

Figure 4: A comparison of TAMP against MLLMs LLaVA-1.5 and Pink on three benchmark. We mark the incorrect part in red, and highlight the correct part in green for comparison.

### 3.6 QUALITATIVE ANALYSIS

As shown in Figure 4, we compare the performance of our model with LLaVA v1.5 and Pink across three multimodal tasks. Our model demonstrates superior fine-grained perception and understanding capabilities in visual question answering, referring expression comprehension, and referring expression generation tasks. In the referring expression comprehension task, LLaVA v1.5 and Pink's coordinate localization clearly deviates to the left, while our model accurately identifies the spatial cue "right side" and generates correct coordinates. In the referring expression generation task, compared to the simple descriptions produced by other models, our model precisely captures fine-grained visual attributes such as clothing colors, fully demonstrating TAMP's superiority in precise spatial localization and fine-grained visual understanding.

## 4 CONCLUSION

This paper proposes the Task-aware Multimodal Pre-Interaction Framework (TAMP), a unified and detector-free LLM for fine-grained downstream tasks. It adequately addresses the challenges of the performance ceiling of additional detectors and domain shift issues, which fundamentally constrain localization accuracy and increase computational complexity. We present a task-aware region connector with a dual-branch architecture to uniformly handle fine-grained visual tasks. The task and semantic intent from instructions are parsed, and task-relevant region features are then highlighted via another processing branch. More importantly, we design an instruction template with a dynamic region placeholder, which is seamlessly replaced with task-aware region features, ensuring integration of region information into text prompts for subsequent multimodal reasoning. With only 595K pretraining data and 797K instruction tuning data, our method achieves state-of-the-art performance on both referring and grounding benchmarks while maintaining strong general VQA capabilities. Our approach successfully injects precise spatial perception into MLLMs without significant computational overhead, establishing a new paradigm for unified fine-grained multimodal understanding.

## REPRODUCIBILITY STATEMENT

To ensure the reproducibility of our work, we have made comprehensive efforts to document all aspects of our implementation and experimental setup. Our complete codebase, including the task-aware region connector architecture, training and evaluation scripts, will be made available as supplementary materials upon acceptance. The model architecture details, particularly the dual-branch design of the task-aware region connector, are fully described in Section 2.2 and illustrated in Figure 2. All experimental settings are specified in Section 3.1 and detailed in Appendix C, including learning rates, batch sizes, and optimization schedules. We use publicly available datasets throughout our experiments: LLaVA-CC3M-Pretrain-595K and Object365 for pretraining, and RefCOCO/RefCOCO+/RefCOCOg, Visual Genome, and standard VQA benchmarks for evaluation, with complete dataset statistics provided in Appendix B. The task-specific instruction templates essential for reproducing our results are comprehensively listed in Appendix D. Our experiments were conducted on 8 NVIDIA A800 GPUs, requiring approximately 48 hours of total training time across three stages. The region-level alignment training details, including the loss functions, are mathematically formulated in Appendix F.

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

## A RELATED WORK

**Multimodal large language models.** With the success of Large Language Models(Achiam et al., 2023; Touvron et al., 2023) in natural language processing, researchers begin to explore ways to extend their powerful reasoning capabilities to multimodal understanding tasks. Early pioneering works on MLLMs primarily focus on aligning visual representations with pre-trained language models. Flamingo(Alayrac et al., 2022) introduces frozen visual encoders paired with Perceiver Resampler to extract visual features, which are subsequently processed by frozen language models through gated cross-attention layers. BLIP-2(Li et al., 2023) proposes Q-Former, bridging frozen image encoders and frozen LLMs to enable efficient vision-language pre-training. Subsequent research has aimed to improve the quality and diversity of multimodal instruction tuning. LLaVA(Liu et al., 2023b) introduces visual instruction tuning by converting image-text pairs into instruction-following formats, enabling MLLMs to follow diverse multimodal instructions. InstructBLIP (Dai et al., 2023) further enhances instruction-aware visual feature extraction through instruction-aware Q-Former. Despite these advances, current alignment mechanisms primarily rely on coarse-grained matching of image-text pairs, making it difficult to establish precise region-language correspondences, which has become a major bottleneck for MLLMs in fine-grained visual perception tasks.

**MLLMs for referring and grounding.** To endow MLLMs with region-level understanding capabilities, researchers have proposed various technical approaches. The first category of methods directly converts bounding box coordinates into text sequences. For instance, works such as Shikra(Chen et al., 2023c), Kosmos-2(Peng et al., 2023), and Pink Xuan et al. (2024) utilize visual localization datasets to enable region-level visual understanding in MLLMs. However, LLMs inherently struggle with processing continuous spatial coordinates, leading to the loss of spatial semantics. The second category introduces specialized expert modules to process region information. GPT4RoI(Zhang et al., 2024b) and PVIT(Chen et al., 2023a) employ RoIAlign at the input end to extract region features, while LLaVA-Grounding(Zhang et al., 2024a) adds a localization decoder at the output end. These methods suffer from task bias: introducing expert modules at the input end benefits referring tasks but provides limited help for localization, while adding modules at the output end has the opposite effect, and significantly increase inference latency. The latest methods utilize pre-trained object detectors (Minderer et al., 2023; Zhu et al., 2020) to provide region proposals. ROD-MLLM(Yin et al., 2025) and Groma(Ma et al., 2024) first generate region proposals and then obtain local features through mechanisms such as ROIAlign(He et al., 2017). While this approach can leverage mature detection technologies, significant computational overhead, detector performance limitations, and a large number of irrelevant candidate boxes also limit the overall effectiveness.

Table 7: The training datasets used for three-stage training.

| Dataset | Stage 1 | Stage 2 | Stage 3 | Data Number |
|---|---|---|---|---|
| LLaVA-CC3M-Pretrain-595K (Liu et al., 2023b) | | ✓ | | 595K |
| LLaVA-Instruct-150K (Liu et al., 2023b) | | | ✓ | 158K |
| Visual Genome(Krishna et al., 2017) | | | ✓ | 108K |
| A-OK-VQA? | | | ✓ | 17K |
| VQAv2(Goyal et al., 2017) | | | ✓ | 83K |
| Flickr30k(Plummer et al., 2015) | | | ✓ | 30K |
| Refcoco (Kazemzadeh et al., 2014) | | | ✓ | 320K |
| Object365(Shao et al., 2019) | ✓ | | | 1M |

Table 8: Summary of the evaluation datasets.

| Task | Dataset | Split | Metric |
|---|---|---|---|
| Visual Question Answering | VQAv2 | test-dev | VQA Score |
| | OK-VQA | val | VQA Score |
| | AOK-VQA | val | VQA Score |
| | VSR | zero-shot test | Accuracy |
| | GQA | test-dev | VQA Score |
| Grounding Task | RefCOCO | val & testA & testB | Accuracy |
| | RefCOCO+ | val & testA & testB | Accuracy |
| | RefCOCOg | val & test | Accuracy |
| Referring Task | Refcocog | val | METEOR & CIDEr |
| | Visual Genome | test | METEOR & CIDEr |

## B  OVERVIEW OF TRAINING AND EVALUATION DATASETS

Table 7 illustrates the dataset usage for the three-stage training strategy. Table 8 lists the benchmark datasets used for model evaluation along with their corresponding evaluation metrics.

## C  MORE IMPLEMENTATION DETAILS

All experiments were conducted on 8 NVIDIA A800 GPUs. The training process consisted of three stages, requiring 20, 2, and 26 hours respectively. The experimental setup involves a three-stage training procedure with distinct hyper-parameter configurations detailed in Table 9.

Table 9: Hyper-parameter for training of different details.

| Configuration | Stage1 | Stage2 | Stage3 |
|---|---|---|---|
| optimizer | AdamW | AdamW | AdamW |
| epochs | 30 | 1 | 6 |
| batch size | 16,384 | 128 | 32 |
| learning rate | 1e-4 | 2e-3 | 5e-4 |
| learning rate schedule | cosine | cosine | cosine |
| warm-up ratio | 0.03 | 0.03 | 0.05 |
| weight decay | 0.0 | 0.0 | 0.02 |
| resolution | 224px | 224px | 224px |

## D  TASK-SPECIFIED INSTRUCTION TEMPLATES

In Section 2.3, we mentioned using task-specific instruction templates to convert various vision-language datasets into instruction-following format for model training. Table 12 provides the complete set of templates

used across different vision-language tasks, including detailed description, visual question answering, region grounding, referring expression, multi-choice VQA, and grounded captioning. These templates incorporate placeholder variables (e.g., <question>, <description>, <location>) that are dynamically filled with task-specific content during training, enabling the model to handle diverse vision-language scenarios in a unified instruction-following paradigm.

# E    MORE EXPERIMENTAL RESULTS

Table 10: Results on the General VQA Benchmark. with Vicuna1.5-7B as the LLM.

| Models | Vision Encoder | Res. | Training Data | | Performance | | | | |
|---|---|---|---|---|---|---|---|---|---|
| | | | #PT Data | #IT Data | VQAv2 | AOK-VQA | VSR | OK-VQA | GQA |
| BaseLine | VIT-L/14 | 336 | 595K | 797K | 79.47 | 60.49 | 67.43 | **59.52** | 65.37 |
| TAMP | VIT-L/14 | 336 | 595K | 797K | **79.55** | **61.07** | **68.66** | 58.73 | **66.88** |

Table 11: Results on the Referring Benchmark with Vicuna1.5-7B as the LLM.

| Method | RefCOCOg | | Visual Genome | |
|---|---|---|---|---|
| | METEOR | CIDEr | METEOR | CIDEr |
| BaseLine | 16.7 | 111.9 | 18.0 | 144.8 |
| TAMP | **17.3** | **114.0** | **19.3** | **162.7** |

The more results that leverage Vicuna1.5-7B(Chiang et al., 2023) as the language model are shown in Table 10 and Table 11. We can observe an improvement in performance on most datasets.

Our comprehensive experimental results in Tables 1, 2, 3, 6, 10, and 11 fully demonstrate the strong generalizability and versatility of the proposed method. Despite the lightweight and simple design of the proposed task-aware region connector , it can be seamlessly integrated into different backbone architectures (CLIP-L, EVA-G paired with LLaMA-2-7B or Vicuna1.5-7B). This unified framework not only significantly enhances the fine-grained perception and understanding capabilities of multimodal large language models, but also maintains their general visual understanding abilities on general VQA tasks. This validates our design philosophy: by extracting key task information from instructions for fine-grained visual tasks and guiding the model to focus on relevant visual regions, effective region-level understanding can be achieved without introducing complex external modules or additional computational overhead.

# F    REGION-LEVEL ALIGNMENT TRAINING DETAILS

To establish region-level vision-language alignment, we specifically train the task-aware region connector. For referring tasks, we employ semantic contrastive loss to ensure $Q_{box}$ accurately captures regional visual content and maps it to the language space. For grounding tasks, we combine contrastive loss with localization loss, enabling the region feature extracted by $Q_{text}$ to align with the text space while containing precise positional information. The specific loss functions are defined as follows.

**Location Loss.** Location Loss combines Smooth L1 loss with Generalized Intersection over Union (GIoU) loss:

$$\mathcal{L}_{location} = \alpha \cdot \mathcal{L}_{smooth\_l1} + (1 - \alpha) \cdot \mathcal{L}_{GIoU}, \tag{4}$$

where the Smooth L1 loss is defined as:

$$\mathcal{L}_{smooth\_l1} = \frac{1}{4} \sum_{i=1}^{4} \begin{cases} 0.5(p_i - t_i)^2, & \text{if } |p_i - t_i| < 1, \\ |p_i - t_i| - 0.5, & \text{otherwise,} \end{cases} \tag{5}$$

where $p_i$ is the predicted value from the model, $t_i$ is the true bounding box value and $i$ is the index that iterates over the bounding box parameters. The GIoU loss takes into account the overall geometric relationship between predicted and ground-truth bounding boxes:

$$\mathcal{L}_{GIoU} = 1 - \left( IoU - \frac{|A_c - U|}{|A_c|} \right), \tag{6}$$

where $A_c$ denotes the area of the smallest enclosing box containing both the predicted and ground-truth boxes, and $U$ represents the area of their union. In our experiments, we set $\alpha = 0.7$ to balance the contributions of both loss components.

**Semantic Contrastive Loss** Semantic Contrastive Loss employs a CLIP-style symmetric contrastive learning loss:

$$\mathcal{L}_{semantic} = \frac{1}{2}(\mathcal{L}_{i2t} + \mathcal{L}_{t2i}), \tag{7}$$

Specifically, for a batch of N samples, we first compute the similarity matrix between normalized features:

$$S_{ij} = \tau \cdot \langle Q_{text_i}, F_{R_i} \rangle, \tag{8}$$

where $F_R$ is a task-aware region feature and $\tau$ is a learnable temperature parameter, initialized to 1/0.07. The similarity of diagonal elements (positive pairs) is then optimized through cross-entropy loss:

$$\mathcal{L}_{i2t} = -\frac{1}{N} \sum_{i=1}^{N} \log \frac{\exp(S_{ii})}{\sum_{j=1}^{N} \exp(S_{ij})}. \tag{9}$$

Table 12: Task-specific instruction templates for various vision-language tasks used in model training.

| Task | Template |
|------|----------|
| Detailed Description | What do you see happening in this image?
What do you think is going on in this snapshot?
What's happening in the scene? |
| Visual Question Answer | I require a brief and clear answer for this question: <question> regarding the image.
I have a question for you: <question> Can you provide a concise answer based on the image ?
Give me a concise answer for <question> while keeping the image in mind. |
| Region Grounding | What are the coordinates of <ref><description></ref> in the image?.
Please locate <ref><description></ref> in the image.
Could you please help me find the coordinates of <ref><description></ref> in the image? |
| Referring - unique description | For the given image, can you provide a unique description of the region <loc> <location> </loc>?
Please generate a distinguishing description for the region <loc> <location> </loc> in the image.
In the photo , how would you describe the selected region <loc> <location> </loc> uniquely? |
| Referring - detailed description | For the given image , can you provide a unique description of the region <loc> <location> </loc>?
Please generate a distinguishing description for the region <loc> <location> </loc> in the image.
In the photo, how would you describe the selected region <loc> <location> </loc> uniquely? |
| Multi-Choices Visual Question Answer | For this image, I want to know which option can answer my question: <question> correctly. The options is <option>.
Referring to the image ¡image¿, please select the answer for this question: <question> from the options <option>.
For this image, I want to know which option can answer my question: <question> correctly. The options is <option>. |
| Grounded Captioning | Can you provide a description of the image and include the coordinates [x1,y1,x2,y2] for each mentioned object?
Tell me about the picture and include position info [x1,y1,x2,y2] for the objects you describe.
Please interpret this image and give coordinates [x1,y1,x2,y2] for each object you mention. |

