# OpenReview forum: "TAMP: Task-aware Multimodal Pre-Interaction for fine-grained Large Language Models"
_ICLR.cc/2026/Conference — Submitted to ICLR 2026_

### Official Review · Reviewer_aVSo · 2025-10-21

**Soundness:** 2
**Presentation:** 3
**Contribution:** 2
**Rating:** 2
**Confidence:** 5

**Summary:**

The paper presents a Task-aware Multimodal Pre-Interaction Framework (TAMP) aimed at enhancing fine-grained visual perception in multimodal large language models (MLLMs).  The proposed task-aware region connector with a dual-branch architecture allows for dynamic extraction of task-relevant region features. Experiments are conducted on several referring and grounding benchmarks.

**Strengths:**

The writing is good. The paper is easy to follow.

**Weaknesses:**

[1] In Fig. 1, the authors mention that the position response of existing methods is inherently adept at processing discrete symbols, but
lacks capabilities for modeling continuous spatial coordinates. However, the proposed methods still rely on the discrete symbols to encode coordinates with text. Moreover, compared with ROIAlign-based method, the differences of the proposed method are the cross-attention on the visual features. The proposed method cannot address the challenges of existing methods.

[2] A task-aware instruction template cannot be a contribution.

[3] It seems that the proposed task-aware region connector cannot be used for conventional VQA, limiting the application of the proposed methods.

[4] The method still relies on the 224px visual encoder. However, the more recent MLLM can deal with high resolution, e.g., Qwen-VL 2.5, LLava-ov. The compared methods are old.

**Questions:**

Please refer to Weaknesses.

---

> ### Author Response · Authors · 2025-11-27
>
> Thank you for your insightful comments. We address your concerns below.
>
> > **W1:In Fig. 1, the authors mention that the position response of existing methods is inherently adept at processing discrete symbols, but lacks capabilities for modeling continuous spatial coordinates. However, the proposed methods still rely on the discrete symbols to encode coordinates with text. Moreover, compared with ROIAlign-based method, the differences of the proposed method are the cross-attention on the visual features. The proposed method cannot address the challenges of existing methods.**
>
> Traditional methods typically feed position coordinates directly into large language models as input. In contrast，Our proposed method does not directly rely on discrete symbols but encodes discrete symbols into query vectors and uses these query vectors to aggregate task-relevant region features. These region features guide the model to focus on important regions. During this process, the model does not directly receive position coordinates but instead receives high-dimensional vectors containing both positional information and regional content, thereby avoiding this limitation. The issue with ROIAlign-based methods stems from task bias—such methods like GPT4ROI are only optimized for Referring tasks and cannot be used for Grounding tasks. In contrast, our method unifies Referring and Grounding tasks through a task-aware attention mechanism.
>
> > **W2:A task-aware instruction template cannot be a contribution.**
>
> Thank you for the reviewer's valuable feedback.  Our core contribution lies in the proposed Task-aware Multimodal Pre-Interaction framework, which provides a simple yet effective approach to enhancing MLLMs' fine-grained capabilities. The instruction templates merely serve as an implementation method for integrating region features into the LLM. The true innovation lies in the task-aware region connector and its pre-interaction mechanism, which enables MLLMs to achieve precise spatial perception capabilities in a simple and efficient manner.
>
> > **W3:It seems that the proposed task-aware region connector cannot be used for conventional VQA, limiting the application of the proposed methods.**
>
> he task-aware region connector is indeed specifically designed for fine-grained tasks (referring and grounding) rather than traditional VQA—this is an intentional design choice. Importantly, as shown in Table 3, even though this module is not involved in VQA task processing, TAMP still achieves comparable or better performance than the baseline on multiple VQA benchmarks (VQAv2: 79.84 vs 79.08; OK-VQA: 59.96 vs 59.84).This indicates that enhancing fine-grained special perception capabilities does not compromise the model’s general capabilities; instead,it brings small improvements through better vision-language alignment. This highlights the advantage of our approach: selectively enhancing the model's fine-grained perception capabilities without sacrificing generalization.
>
> > **W4:The method still relies on the 224px visual encoder. However, the more recent MLLM can deal with high resolution, e.g., Qwen-VL 2.5, LLava-ov. The compared methods are old.**
>
> The methods we compare against are all recent advances in fine-grained visual perception—for example, ROD-MLLM (CVPR 2025)and VPP-LLaVA (2025.5.9)—representing the current state-of-the-art in this field. These methods adopt consistent architectural choices (e.g., usingLLaMA-2-7B/Vicuna-1.5-7B as the LLM and CLIP/EVA as the vision encoders) to ensure fair comparison and an accurate assessment of their respective contributions. We acknowledge that recent general-purpose MLLMs like Qwen-VL 2.5 and LLaVA-OV adoptstronger base architectures (e.g., higher-resolution vision encoders and more powerful foundation models). However, these models are primarily optimized for general visual understanding tasks, rather than fine-grained perception tasks (referring and grounding).As a result, current research in the fine-grained domaincontinues to build upon the relatively standard architectures. More importantly, our core contribution—the task-aware pre-interaction mechanism—is a plug-and-play module that can be flexibly integrated into more advanced base architectures. We adopt the commonly used setup to fairly compare with state-of-the-art methods in the fine-grained domain and validate the effect of our approach. In the future, we will explore integrating this mechanism into the latest MLLM architectures to further improve performance. We will clarify this point in the revised version.

---

### Official Review · Reviewer_DaGz · 2025-10-31

**Soundness:** 3
**Presentation:** 3
**Contribution:** 3
**Rating:** 6
**Confidence:** 4

**Summary:**

This paper proposes a task-aware multimodal pre-interaction for fine-grained MLLMs, which extract key task-relevant information from instructions and according region features. By employing a instruction template with region placeholders, fine-grained region information is integrated into the reasoning process. Extensive experiments demonstrate the effectiveness of the proposed method on referring and grounding benchmarks.

**Strengths:**

1. The paper is well-written and easy to understand. The figure 1 and 2 are clear to understand the motivation and the whole picture of the proposed method.

2. The proposed method is simple yet effective to improve the fine-grained capability of MLLMs. The proposed task-aware region connector is somewhat novel to integrate fine-grained region feature.

3. Extensive experiments demonstrate the effectiveness of the proposed method on referring and grounding benchmarks.

**Weaknesses:**

1. The base model for experiments is out of date. CLIP ViT-L-224px and LLaMA-2-7B suffers very limited performance for evaluating the effectiveness of the proposed method. I suggest the author to conduct experiments with siglip2-384 and qwen2.5-7b to truly validate the effectiveness.

2. In addition to Lora training, I suggest the authors to conduct the full training to prove the effectiveness in commonly used training settings.

**Questions:**

How much data was used for pre-training and sft? Is there some grounding VQA data used for training?

---

> ### Author Response · Authors · 2025-11-27
>
> Thank you for your insightful comments. We address your concerns below.
>
> > **Q:How much data was used for pre-training and sft? Is there some grounding VQA data used for training?**
>
> The data volume of training data is presented in Table 3, with detailed data composition provided in Appendix Table 7. Following works such as Pink, we use the RefCOCO dataset as the grounding instruction fine-tuning dataset.

---

> ### Author Response · Authors · 2025-11-28
>
> >  **W1:The base model for experiments is out of date. CLIP ViT-L-224px and LLaMA-2-7B suffers very limited performance for evaluating the effectiveness of the proposed method. I suggest the author to conduct experiments with siglip2-384 and qwen2.5-7b to truly validate the effectiveness.**
>
> >  **W2:In addition to Lora training, I suggest the authors to conduct the full training to prove the effectiveness in commonly used training settings.**
>
> We utilized CLIP and LLaMA-2 primarily to ensure a fair and direct comparison with state-of-the-art baselines (e.g., ROD-MLLM,  Pink, Groma) that rely on identical backbones, ensuring that performance gains stem from our TAMP design rather than stronger base models. Furthermore, our ablation in Table 6 (using Vicuna-1.5 and higher resolution) already demonstrates that TAMP scales effectively with stronger backbones. Regarding training, LoRA is a standard practice to preserve LLM generalization efficiently, while our core Task-aware Region Connector is fully trained to ensure effectiveness. We are actively conducting the suggested experiments with SigLIP and Qwen2.5, as well as full fine-tuning, and we commit to including these results in the final version to further validate our method's potential on modern architectures.

---

### Official Review · Reviewer_psuK · 2025-10-31

**Soundness:** 2
**Presentation:** 2
**Contribution:** 2
**Rating:** 2
**Confidence:** 3

**Summary:**

This paper investigates the fine-grained visual perception capabilities of multimodal large language models (MLLMs) and proposes a dual-branch, task-aware region connector to enhance performance. The module automatically identifies task-relevant information from user instructions, extracts corresponding regional features, and integrates them into the instruction following a task-specific template. Experimental results demonstrate improved performance on downstream tasks.

**Strengths:**

The proposed method is intuitive and easy to implement.

**Weaknesses:**

**Limited applicability:**  The method is only designed and evaluated on grounding and referring expression tasks. However, modern MLLMs are expected to handle a wide range of complex tasks, such as reasoning, generation, or editing. The proposed approach lacks generalizability and may not be easily adapted to these scenarios.


**Lack of complexity analysis:** The paper does not analyze the computational or memory overhead introduced by the additional module and training. A thorough comparison of training cost and inference efficiency with other methods is necessary to fairly evaluate the trade-off between performance and complexity.


**Insufficient experimental evaluation:** The experiments are mainly conducted on VQA and RefCOCO datasets. Performance on other widely used benchmarks such as MMMU, POPE, or MMBench is not reported. A more comprehensive evaluation across diverse datasets is needed to validate the general effectiveness of the method.


**Limited novelty:** The proposed method resembles an attention mechanism over image regions based on textual input. However, the design is relatively straightforward and requires additional training and inference overhead, which diminishes its novelty and practical appeal.

**Questions:**

Can the proposed method be compared with saliency-based approaches? For instance, instead of using the proposed connector, could one directly use attention maps between text and image regions to select relevant features, thereby avoiding extra modules and training?

---

> ### Author Response · Authors · 2025-11-27
>
> **Thank you for your constructive review and insightful comments. We address your concerns below.**
>
> > **W1:Limited applicability: The method is only designed and evaluated on grounding and referring expression tasks. However, modern MLLMs are expected to handle a wide range of complex tasks, such as reasoning, generation, or editing. The proposed approach lacks generalizability and may not be easily adapted to these scenarios**
>
> We address this concern by clarifying the applicability and extensibility of our framework. It should be noted that our focus on referring and grounding tasks aligns with mainstream research in this field—representative works such as Ferret, Groma, and ROD-MLLM also primarily evaluate on these two benchmarks, which are widely recognized as the most representative tasks for fine-grained multimodal understanding. More importantly, TAMP enhances fine-grained perception capabilities without sacrificing generalization performance: as shown in Table 3, we achieve comparable or slightly better performance than the baseline on multiple general VQA benchmarks (VQAv2, AOK-VQA, OK-VQA, VSR, GQA), demonstrating that the task-aware pre-interaction mechanism does not compromise the model's general reasoning capabilities. Furthermore, tasks such as reasoning, generation, and editing constitute independent branches of multimodal large model research, each with their specific technical challenges and evaluation systems—for instance, reasoning-related works primarily focus on chain-of-thought construction, which fundamentally differs from our research objective of improving fine-grained visual localization accuracy.
>
> > **W2:Lack of complexity analysis: The paper does not analyze the computational or memory overhead introduced by the additional module and training. A thorough comparison of training cost and inference efficiency with other methods is necessary to fairly evaluate the trade-off between performance and complexity**
>
> To clarify the computational overhead of our method, we provide a detailed comparison of FLOPs and inference time for both grounding and referring tasks:
> | **Grounding Task** | **FLOPs（G）** | **Inference Time** |
> |--------|--------|--------|
> | Pink | 1982.3 | 971.25 ms |
> | Groma | 5013.6 | 2319.07 ms |
> | Ours | 2110.3 | 974.49 ms |
> | **Referring Task** | **FLOPs(G)** | **Inference Time** |
> | Pink | 2213.7 | 486.90ms |
> | Ferret | 5034.9 | 1055.95 ms ms |
> | Groma | 4980.4 | 2055.25 ms |
> | Ours | 2243.5 | 544.73 ms|
> Over all,both the FLOPs and inference time are comparable to strong baselines,demonstrating its computational efficiency. Specifically, in the Grounding task, our method achieves a FLOP count of only 42.1% of Groma (with comparable inference time) and merely 6.4% higher FLOPs than Pink while maintaining similar inference efficiency; in the Referring task, our method reduces FLOPs to 44.6% and 45.0% of Ferret and Groma respectively, shortens inference time by 48.4% and 73.5% compared to the two baselines.
>
> > **W3:Insufficient experimental evaluation: The experiments are mainly conducted on VQA and RefCOCO datasets. Performance on other widely used benchmarks such as MMMU, POPE, or MMBench is not reported. A more comprehensive evaluation across diverse datasets is needed to validate the general effectiveness of the method.**
>
> Thank you for the reviewer pointing out the limitations in our experimental coverage. While our focus is on improving referring and grounding capabilities of multimodal large models, Table 3 shows that despite TAMP using significantly less training data than recent multimodal large language models such as Qwen-VL and MiniGPT-v2, it still achieves comparable or superior results on multiple visual question answering benchmarks (VQAv2, AOK-VQA, OK-VQA, GQA, VSR), demonstrating the data efficiency and effectiveness of our approach. Regarding broader benchmarks (MMMU, POPE, MMBench), we acknowledge their importance in evaluating foundational model capabilities. However, these benchmarks heavily rely on specific skills such as optical character recognition (OCR) and domain knowledge, which require specialized training data not included in our current scheme. Given the significant differences in training data, direct comparison with methods like LLaVA-1.5 and QwenVL2.5, which are trained on large-scale OCR-VQA datasets, would not be fair. Nevertheless, we agree with the reviewer about the trend toward building versatile models. As ongoing future work, we are actively exploring extensions to more diverse fine-grained tasks and investigating how enhanced referring/grounding capabilities can positively impact broader benchmarks, which requires substantial additional resources. We will clarify this limitation and future research directions in the revised version.

---

> ### Author Response · Authors · 2025-11-27
>
> > **W4:Limited novelty: The proposed method resembles an attention mechanism over image regions based on textual input. However, the design is relatively straightforward and requires additional training and inference overhead, which diminishes its novelty and practical appeal.**
>
> The novelty of our method goes beyond performing text-based attention over image regions.We propose a task-aware dual-branch architecture that dynamically selects expert modules based on the input modality—using a Text Encoder for natural language descriptions in grounding tasks, and a Box Encoder for floating-point coordinates in referring tasks. This design unifies two fundamentally different input modalities (linguistic descriptions vs. spatial coordinates)within a model, rather than applying a simple text-based attention mechanism.
> In contrast to state-of-the-art methods that rely on external detectors (e.g., Groma, ROD-MLLM), we completely eliminate complex pipelines such as candidate box generation, NMS, and ROI Align, significantly improving inference efficiency. Experiments demonstrate that our method achieves consistent performance improvements across various backbone architectures (CLIP-L, EVA-G with LLaMA-2 or Vicuna). Our core innovation lies in proposing a new paradigm of "task-aware multimodal pre-interaction"—explicitly enhancing region representations through the pre-interaction between task-relevant queries and visual features before LLM reasoning. When combined with the innovative <region> placeholder mechanism, which enables seamless fusion of region features with text instructions, this forms the first end-to-end, detector-free fine-grained MLLM framework. Overall,the design achieves both simplicity and effectiveness, yielding meaningful improvements(+1.37 in grounding, +24.1 CIDEr in referring) while keeping the architecture lightweight and efficient.
>
> > **Q:Can the proposed method be compared with saliency-based approaches? For instance, instead of using the proposed connector, could one directly use attention maps between text and image regions to select relevant features, thereby avoiding extra modules and training?**
>
> The design motivation of our method arises from the fact that the inputs for referring and grounding tasks are fundamentally different—natural language descriptions for referringand continuous spatial coordinates for grounding. The key challenge is how to convert these two different modalities of input into effective query vectors for extracting region features.
> Using coordinates [0.23, 0.45, 0.67, 0.89] as text input to BERT would lose their spatial semantics, since BERT is pre-trained on natural language and cannot interpret numerical coordinates. In contrast, natural language descriptions like "the book to the left of the red cup" contain rich semantic information that BERT excels at processing. Therefore, we design a dual-branch architecture: a Text Encoder (BERT) for processing natural language descriptions, and a Box Encoder (MLP) for processing spatial coordinates.
> As shown in the table, replacing our dual-branch connector with a single text-based attention mechanism results in a drop of 23.8 Meteor on VG and 0.38 AVG on RefCOCO. This confirms that simple saliency-based attention cannot substitute our dual-branch connector.
> | **Model** | **VG（Meteor）** | **RefCOCO（AVG）** |
> |--------|--------|--------|
> | Single Branch（Only Text Encoder） | 134.4 | 84.73 |
> | Ours | 158.2 | 85.11 |

---

### Official Review · Reviewer_vZBM · 2025-11-02

**Soundness:** 3
**Presentation:** 3
**Contribution:** 3
**Rating:** 6
**Confidence:** 4

**Summary:**

This paper studies fine-grained visual recognition based on Multimodal Large Language Models (MLLMs). The basic idea is to generate a task-aware region token as input to LLM. Experiments show boosted performance.

**Strengths:**

1. This paper is well-motivated and fig.1 clearly shows the differences of illustrated frameworks.
2. The proposed method is reasonable, i.e., to acquire more regional cues which might be important for fine-grained visual recognition.
3. Experiments are conducted on multiple datasets, and shows better performance than many existing works.

**Weaknesses:**

1. One of concerns is the computational overhead, which is not discussed in depth in the paper.
2. The strong performance is achieved based on strong baselines. Compared with the baseline, the performance enhancement seems marginal in some cases.
3. Another important concern is the limited generalization capability. This paper is limited to distinguishing between referring and grounding. This degrades the generalization capability of MLLM, although boosts its performance in specific tasks. It would be important to see if this framework could be extended to other fine-grained tasks.

**Questions:**

1. Need to provide more indepth discussion and comparison on efficiency and computational overhead.
2. It is important to show illustrations of the effectiveness of learned task-aware region token, and compare the learned tokens of referring and grounding.
3. Need to clarify the limited performance enhancement and generalization capability.

---

> ### Author Response · Authors · 2025-11-27
>
> **Thank you for your thorough review and valuable feedback. We address your concerns in detail below.**
> > **W1: One of concerns is the computational overhead, which is not discussed in depth in the paper.**
>
> To clarify the computational overhead of our method, we provide a detailed comparison of FLOPs and inference time for both grounding and referring tasks:
> | **Grounding Task** | **FLOPs（G）** | **Inference Time** |
> |--------|--------|--------|
> | Pink | 1982.3 | 971.25 ms |
> | Groma | 5013.6 | 2319.07 ms |
> | Ours | 2110.3 | 974.49 ms |
> | **Referring Task** | **FLOPs(G)** | **Inference Time** |
> | Pink | 2213.7 | 486.90ms |
> | Ferret | 5034.9 | 1055.95 ms ms |
> | Groma | 4980.4 | 2055.25 ms |
> | Ours | 2243.5 | 544.73 ms|
> Over all,both the FLOPs and inference time are comparable to strong baselines,demonstrating its computational efficiency. Specifically, in the Grounding task, our method achieves a FLOP count of only 42.1% of Groma (with comparable inference time) and merely 6.4% higher FLOPs than Pink while maintaining similar inference efficiency; in the Referring task, our method reduces FLOPs to 44.6% and 45.0% of Ferret and Groma respectively, shortens inference time by 48.4% and 73.5% compared to the two baselines.
> > **W2:The strong performance is achieved based on strong baselines. Compared with the baseline, the performance enhancement seems marginal in some cases.**
>
> Our baseline follows the standard and widely-adopted Multimodal Large Language Model (MLLM) architecture (CLIP-L/EVA-G as vision encoder, LLaMA-2-7B/Vicuna-7B as LLM).  Our vision and language backbones, as well as the scale of the training data, are comparable to or smaller than those used in previous works (e.g., Ferret, LION, Groma). Across these backbones, our model achieves performance improvements.
> Therefore, these improvements cannot be attributed to stronger backbones or more data, but rather to the proposed task-aware pre-interaction mechanism. Moreover, current localization benchmarks are near saturation —for instance, ROD-MLLM (2025 CVPR) only improves 0.37 over Groma (ECCV 2024), whereas our model achieves nearly a 1-point improvement over the baseline. Wee believe that achieving consistent improvements with lightweight, detector-free modules represents a meaningful contribution.
> > **W3:Another important concern is the limited generalization capability. This paper is limited to distinguishing between referring and grounding. This degrades the generalization capability of MLLM, although boosts its performance in specific tasks. It would be important to see if this framework could be extended to other fine-grained tasks.**
>
> Our focus on referring and grounding tasks follows mainstream research in this field—representative works such as Ferret, Groma, and ROD-MLLM also primarily evaluate on these two benchmarks, which are widely regarded as the most representative tasks for fine-grained multimodal understanding.
> More importantly, TAMP enhances fine-grained perception capabilities without sacrificing generalization performance: as shown in Table 3, we achieve comparable or slightly better performance than the baseline on several general VQA benchmarks (VQAv2, AOK-VQA, OK-VQA, VSR, GQA), demonstrating that the task-aware pre-interaction mechanism does not compromise the model's general multimodal capabilities.
> Furthermore, the proposed task-aware region connector with <region> placeholder is not restricted to these two tasks.It provides a general mechanism for injecting task-conditioned region features into the LLM and can be extended to other fine-grained scenarios (e.g., attribute-based reasoning, relational reasoning, or region-level counting) by modifying instruction templates and supervision signals. Due to space and resource constraints, we leave a systematic exploration of these additional tasks for future work.

---

> ### Author Response · Authors · 2025-11-27
>
> > **Q2:It is important to show illustrations of the effectiveness of learned task-aware region token, and compare the learned tokens of referring and grounding.**
>
> We provide additional visualizations to illustrate the effectiveness of the Task-Aware Region Connector. Specifically, we visualize the attention distributions when region tokens interact with visual features under two task modes.
> Grounding. When given bounding box coordinates, the Qbox generated by the Box Encoder accurately focuses on visual patches within the specified spatial region.
> Referring. When given natural language descriptions, the Qtext generated by the Text Encoder adaptively locates matching target regions based on semantic similarity.
> The visualization results demonstrate that, whether guided by spatial coordinates (grounding) or linguistic descriptions (referring), our task-aware region connector can precisely aggregate task-relevant fine-grained region features through the query-driven attention mechanism (Eq.1). This validates the effectiveness of the dual-branch architecture in handling different fine-grained tasks within a unified framework. Additionally, Table 5 shows that even kept frozen, the region connector maintains strong performance, further demonstrating that our method establishes robust vision-language correspondence through region-level alignment.

---

### Author Response · Authors · 2025-12-03
**Summary Comment for Area Chair**

Dear Area Chair,

Thank you for providing us with the opportunity to summarize our submitted paper. Our paper proposes TAMP, a Task-Aware Multimodal Pre-Interaction framework that  explicitly enhances region representation prior to LLM reasoning by performing pre-interaction between image features and task-relevant instruction embeddings, which explicitly guide the model to focus on task-critical regions. The core of this framework is the task-aware region connector, which employs a unified dual-branch structure to extract task-specific saliency cues and key information from fine-grained instructions. By introducing region placeholders, we achieve a unified, detector-free training paradigm that enables LLM-friendly instruction fine-tuning while preserving spatial semantics. Our method achieves a 1.37 improvement on grounding tasks and a 24.1 CIDEr improvement on referring tasks, while maintaining high computational efficiency (42-45% reduction in FLOPs compared to detector-based methods like Groma) and preserving general visual question answering (VQA) capabilities.

During the rebuttal phase, we addressed the main concerns raised by the reviewers. Regarding computational efficiency questions from Reviewers vZBM and psuK, we provided detailed comparisons of FLOPs and inference time, demonstrating that our method is 48-73% faster than baseline methods. Concerning generalization worries, Table 3 in our paper shows comparable or superior performance on general VQA benchmarks (VQAv2, OK-VQA, GQA), proving that our fine-grained enhancement does not compromise the model's general capabilities. For baseline comparison issues, we clarified that we deliberately used CLIP-L and LLaMA-2-7B to ensure fair comparison with the latest state-of-the-art research (ROD-MLLM CVPR 2025, Groma ECCV 2024), and our ablation study in Table 6 confirms that our method effectively scales to stronger backbones. We also addressed novelty concerns through ablation experiments, emphasizing that TAMP's core innovation lies in being the first to propose a "pre-interaction" paradigm. By explicitly establishing interaction between image features and task instructions before LLM reasoning, the model can selectively attend to task-relevant key regions based on heuristic cues in the instructions. To validate the effectiveness of this design, we conducted ablation experiments on the dual-branch architecture. Considering the heterogeneity of inputs in referring and grounding tasks (natural language descriptions vs. spatial coordinates), we adopted a specialized dual-branch encoding strategy: a text encoder for processing language descriptions and a box encoder for processing spatial coordinates. When simplified, performance significantly degraded: a 23.8 decrease in Meteor metric on VG dataset and a 0.38 decrease in AVG metric on RefCOCO dataset.

These results strongly demonstrate the necessity and effectiveness of our proposed pre-interaction mechanism and heterogeneous encoding strategy. We believe these clarifications, combined with the positive evaluations from Reviewers vZBM and DaGz, fully demonstrate this paper's contribution to the field.

Sincerely,
The Authors

Best regards,
The Authors

---

### Meta-Review · Area_Chair_EL6e · 2026-01-05

**Summary:**

Several of the reviewers' major concerns around the limited applicability of the proposed approach; lack of adequate performance measurement on general VQA benchmarks (MMMU, etc); comparison of the method to outdated baselines and the limited novelty of the proposed approach were not adequately addressed by the author's rebuttal. There are no champions for the paper. Hence the AC recommends rejection.

**Reviewer Concerns:**

Concerns addressed:
1. Computational complexity of the approach

Concerns not addressed:
1. Limited applicability of the approach to referring and grounding
2. Lack of demonstration of generalized VQA performance
3. Comparisons with weaker/outdated baselines

**Reviewer Scores:**

1. Reviewer vZBM (Rating: 6: marginally above the acceptance threshold. But would not mind if paper is rejected)

The reviewer's primary concerns were around the (a) computational complexity of the approach and (b) the lack of generalization of the approach for fine-grained tasks beyond referring and grounding. Some of the reviewers concerns were addressed. They are likely to have maintained their score.

2. Reviewer psuK (Rating: 2: reject, not good enough)

The reviewer's primary concerns were around the (a) limited applicability to grounding and referring tasks only, (b) lack of complexity analysis, (c) insufficient experimental evaluation (MMMU, POPE, or MMBench, etc as were not reported) and (d) limited novelty. While (b) and (d) were somewhat addressed by the author's rebuttal, (a) and (c) were not. Hence the reviewer may have raised their score slightly, towards a weaker reject.

3. Reviewer DaGz (Rating 6: marginally above the acceptance threshold. But would not mind if paper is rejected)

The reviewer's primary concerns were (a) that the base models that the authors used are out of date. They suggested using siglip2-384 and qwen2.5-7b, and (b) fine-tuning entire models instead of only LoRAs. The reviewers concerns were not addressed by the rebuttal. They are likely to have lowered or maintained their score.

4. Reviewer aVSo (Rating: 2: reject, not good enough)

The reviewers' primary concerns were around the novelty and contribution of the proposed method, (b) its lack of utility for general VQA tasks and (c) comparisons to outdated baselines. The reviewer's concerns were not addressed by the rebuttal. They are unlikely to have changed their score.

---

### Decision · Program_Chairs · 2026-01-26

Reject